# The Role of High Triglycerides Level in Predicting Cognitive Impairment: A Review of Current Evidence

**DOI:** 10.3390/nu13062118

**Published:** 2021-06-20

**Authors:** Alina Mihaela Dimache, Delia Lidia Șalaru, Radu Sascău, Cristian Stătescu

**Affiliations:** 1Neurology Outpatient Clinic, Department of Chronic Diseases, Hospital of Chronic Diseases Târgu Frumos, 705300 Iași, Romania; alinadimache1@gmail.com; 2Faculty of Medicine, University of Medicine and Pharmacy Grigore T. Popa, 700115 Iași, Romania; radu.sascau@gmail.com (R.S.); cstatescu@gmail.com (C.S.); 3Institute of Cardiovascular Diseases, 700503 Iasi, Romania

**Keywords:** triglycerides, cognitive decline, dementia, Alzheimer’s

## Abstract

The burden of cognitive disorders is huge and still growing, however the etiology and the degree of cognitive impairment vary considerably. Neurodegenerative and vascular mechanisms were most frequently assessed in patients with dementia. Recent studies have shown the possible involvement of triglycerides levels in cognitive function through putative mechanisms such as brain blood barrier dysfunction or amyloid metabolism imbalance, but not all research in the field found this association. Several clinical studies evaluated the relationship between different forms of cognitive decline and levels of serum triglycerides, independent of other cardiovascular risk factors. This review focuses on the role of triglycerides in cognitive decline, cerebral amyloidosis and vascular impairment. Considering that the management of hypertriglyceridemia benefits from lifestyle modification, diet, and specific drug therapy, future studies are requested to appraise the triglycerides–cognitive impairment relationship.

## 1. Introduction

The rise of life expectancy contributes to the burden of the aging-dependent decline of physical and mental functions. One of the most common and challenging neuropsychological condition is cognitive impairment (CI), defined as the decrease of intellectual functions ranging from mild forms of forgetfulness to severe and debilitating dementia [1]. With a prevalence of around 50 million people, dementia is recognized by the World Health Organization as a public health priority, and this number is expected to double every 20 years [2]. Two of the commonest subtypes in order of frequency are Alzheimer’s Disease and vascular dementia, but it is difficult, particularly in epidemiological studies, to establish an accurate subtype diagnosis. [3].

The existing evidence shows a link between vascular risk factors and the development of vascular cognitive impairment [4,5,6]. The accumulated findings of a causal role of dyslipidemia in the etiology of neurodegeneration and cognitive decline are strong, but the results are still debatable [7,8,9,10].

This review will focus on the possible relationship between triglycerides, cognitive decline, cerebral amyloidosis, and vascular impairment.

## 2. Clinical and Physiopathological Classification of Cognitive Impairment

According to the fifth edition of the Diagnostic and Statistical Manual of Mental Disorders (DSM) published by the American Psychiatric Association in 2013, mild cognitive impairment corresponds to minor neurocognitive disorders, while dementia corresponds to major neurocognitive disorders, respectively [5,11].

In the trajectory from normal cognitive function to dementia, mild cognitive impairment (MCI) is an intermediate stage or a pre-dementia stage. MCI is commonly used for classifying individuals that present cognitive decline but do not meet the criteria for Alzheimer’s disease [12]. Although Alzheimer’s disease is the most commonly diagnosed cause of cognitive dysfunction among the elderly, cognitive impairment caused by vascular disease, including subclinical brain injury, silent brain infarction (SBI), and clinically overt stroke are listed among the independent contributors to cognitive dysfunction [4]. Alzheimer’s disease (AD) is seen as a continuum from preclinical, asymptomatic stage to mild cognitive impairment and dementia stage [13]. The preclinical phase used in research settings (interventional and observational studies) is based on biological criteria where the explorations for β amyloid and tau protein are positive, with or without the presence of neurodegeneration, but in the absence of symptoms [14].

However, not all individuals with MCI will develop dementia, most studies showing a progression rate of 10–15% per year [15]. MCI requires the objective presence of deterioration in at least one cognitive domain such as memory, executive functions, attention, language, or visuospatial skill. The key criteria used to distinguish MCI from the dementia stage are the conservation of daily functional abilities and the absence of noteworthy impairment in social and occupational functioning [11]. According to the literature, 10–20% of the population aged 60 years and older develop MCI [11,15]. Both MCI and dementia are increasing with age [16]. Better awareness and understanding of MCI and the other stages of cognitive impairment may contribute to the development of better diagnostic mechanisms and consequently to the development of therapeutic and non-therapeutic interventions for MCI as early as possible.

From the pathological perspective, there is a dispute about the role of various types of vascular lesions that contribute to cognitive impairment, namely large cortical infarcts, lacunar infarcts, subcortical white matter disease, subcortical infarcts, or a combination of these. The clinical characterization of vascular dementia is particularly difficult because the Alzheimer disease clinical syndrome can begin after a stroke, as well as patients with Alzheimer disease symptomatology can have strokes during the disease. Furthermore, vascular lesions can lower the threshold for the clinical manifestations of Alzheimer’s disease. An early study investigated the relationship of brain infarction to the clinical expression of AD [17]. Among 61 participants who met the neuropathological criteria for AD, those with brain infarcts had a poorer cognitive function and a higher prevalence of dementia than those without infarcts. Participants with lacunar infarcts in the basal ganglia, thalamus, or deep white matter had an especially high prevalence of dementia, compared with those without infarcts. Among all 102 participants, atherosclerosis of the circle of Willis was strongly associated with lacunar and large brain infarcts [17]. These results suggest that cerebrovascular disease is closely related to the different forms of cognitive impairment.

A large number of studies have reported strong relationships between indices of vascular aging and either cognitive impairment or silent cerebral small-vessel disease. The fact that these relationships were independent of age and classic cardiovascular risk factors suggests common pathophysiological mechanisms linking large-artery damage to cerebral small-vessel disease.

Hypercholesterolemia is well recognized as a major risk factor for atherosclerotic cardiovascular disease. Nevertheless, it was observed that after optimal treatment and the achievement of lower LDL-cholesterol levels, there is still a considerable residual risk [18]. High levels of triglycerides are associated with atherosclerotic cardiovascular disease risk [19,20,21]. The supposed mechanisms incriminated for the increased atherogenicity of triglycerides include endothelial dysfunction, foam cell formation, inflammation, and cytokines regulation [22].

## 3. Triglycerides and Different Types of Cognitive Impairment

### 3.1. Triglycerides and Cognitive Decline

Triglycerides (TG) are simple lipids involved in the storage and transport of energy. There are two sources for serum TG: gut absorption and liver synthesis. Depending on three fatty acids and their combinations, there are more than 6000 species of TG [23]. These lipids are an important part of triglyceride-rich proteins (TRLs) which comprise very-low-density lipoprotein (VLDL), chylomicrons, and the particles that remain after their catabolism [20,22].

Banks et al. detected TG in human cerebrospinal fluid and moreover, they found that radioactive TG triolein crosses the blood-brain barrier (BBB) in mice. The authors have demonstrated the improvement of cognition after gemfibrozil administration with a subsequent decrease of serum TG level [24]. However, the evidence regarding the passage of the BBB by the TG is very scarce and requires more studies. Data from animal studies suggest that TG contribute to cognitive decline through the impaired maintenance of the *N*-methyl-d-aspartate component of hippocampal long-term potentiation, and consider that lowering TG could reverse the cognitive impairment in mice [25].

Data from several longitudinal studies have shown a correlation between serum TG in midlife and the risk of cognitive impairment in the elderly [7,26]. Increased TG level in middle age years was associated with the risk of dementia 25 years later in a large cohort including Japanese-American men in the Honolulu–Asia Aging Study [26]. A recent longitudinal cohort study with 13,997 eligible participants pointed out the association of midlife high total cholesterol and TG levels with advanced cognitive decline assessed by memory, executive function, sustained attention, and processing speed test, after 20 years of follow-up [7]. Reynolds et al. found that lower TG levels in women before 65 years (average age at baseline for lipid evaluation was 63.76 years) were associated with and were also predictive for future better memory performance and verbal abilities in a 16-year longitudinal study with 819 participants. The TG concentrations were less predictive in men [27]. Ancelin et al. outlined a gender-dependent outcome of TG levels and incidence of cognitive impairment risk in a 7-year follow-up study with 7053 participants: 4308 women and 2745 men, mean age 73.9 and 73.7 years, respectively. In men without cardiovascular diseases increased TG levels were associated with a significant incidence of all-cause dementia, except for AD. Interestingly, in women, both low and high TG were correlated with decreased risk of AD, independent of apolipoprotein E (APOE) status and vascular factors. The authors suggested that the association of high TG level and low risk of AD could be restricted to those persons carrying the AA polymorphism of apolipoprotein A5 (APOA5) [28].

A few years earlier, Henderson et al. found no association between TG level and memory in a longitudinal study performed on 326 women (aged 52–63 years) during 8 years of follow-up [29]. Moreover, a prospective study with 5 years follow-up indicated an association between increased TG levels and a lower risk of cognitive decline in 930 Chinese subjects with a mean age of 94 years [30].

The association of higher TG levels with poorer cognitive function is also supported by several cross-sectional studies. The outcomes from a case-control Chinese study with 112 MCI subjects and 115 cognitively normal participants aged >65 years indicated a correlation between high TG level and MCI [31]. Data from another cross-sectional study performed on 121 African-American participants (mean age 43.74 years) revealed inferior results of verbal learning performance on the California Verbal Learning Test II (CVLT) in subjects with increased TG levels, without any relation to blood pressure [32]. In addition, higher levels of serum TG were significantly associated with poorer executive function, but not with memory, independent of other vascular risk factors, apolipoprotein E4 (ApoE4) status, and cerebral white matter microstructure. 251 non-demented elderly have participated in this cross-sectional study, mean age 78 years, 54% male [33].

In contrast, the results from a Chinese cross-sectional study performed on 836 subjects (majority females) indicated the association of high normal plasma TG with better cognitive function in elderly over 80 years [34]. Increased TG were also negatively associated with cognitive impairment in male aged 40–55 years in a cross-sectional study with 1762 participants [35]. The results from the cross-sectional Leiden 85-Plus Study showed no correlation of the TG concentrations with cognitive decline in 561 subjects aged ≥85 years [36]. Furthermore, in males, recent data from a cross-sectional study on 2150 Japanese participants (aged 60–90 years) found that high TG levels decrease the global cognitive impairment risk [37].

A recent cross-sectional study with 689 participants from the Alzheimer’s Disease Neuroimaging Initiative cohort (160 with AD, 339 with MCI, and 190 cognitively normal) revealed that reduced levels of two long-chain, polyunsaturated fatty acid-containing TG (PUTG), principal component 3 (PC3) and principal component 5 (PC5), respectively, were associated with lower cognitive performance, hippocampal and entorhinal cortical atrophy [23].

Table 1 summarizes the included studies about triglycerides and cognitive decline.

### 3.2. Triglycerides and Cerebral Amyloidosis

The etiology of AD remains unclear, but evidence suggests that the major neuropathological hallmark of AD is the accumulation of amyloid protein in senile plaques due to over-production or impaired clearance of β-amyloid (Aβ) peptides and the deposition of neurofibrillary tangles (NFTs), which give rise to synaptic loss and neurodegeneration [38,39]. It has been proposed that asymptomatic cerebral amyloidosis should be the first stage of preclinical AD for research purposes. On the other hand, brain Aβ deposition is noticed in a substantial percentage of elderly without cognitive impairment [40]. Not all individuals with brain Aβ deposition will exhibit dementia symptoms during their life. A possible protective role of genetic factors, brain reserve, or environmental factors is taken into consideration [41]. The biomarkers of brain Aβ plaque load are positron emission tomography (PET) Aβ imaging and low cerebrospinal fluid (CSF) Aβ42 [14]. Nevertheless, these markers appear to assess the fibrillar forms of Aβ but not the oligomeric ones which seem to be critical for synaptic impairment [41].

Regarding the dynamic of brain amyloid-beta deposition, there is a sigmoid shape of the evolution with an exponential phase that corresponds to normal cognitive status, and a plateau phase which is reached before the appearance of clinical symptoms or to atrophy on MRI [14]. Considering that cerebral amyloid accumulation is starting at least 10 years before late MCI and AD dementia development, a cross-sectional study might have difficulties identifying contributing factors to cerebral amyloidosis in participants with cognitive deterioration [42]. A recent study performed on 942 elderly individuals (average age 79.7 years) in the Mayo Clinic Study of Aging revealed an association between midlife dyslipidemia (total cholesterol) and amyloid β brain deposition [43]. Nägga et al. support the idea that risk factors related to early AD changes should be investigated in normal cognitive participants [44].

Data from preclinical studies revealed that in AD mouse models with abundant plasma Aβ, VLDL (very low-density lipoprotein) TG levels precede amyloid brain deposition [45].

In humans, a cross-sectional study showed that high serum TG level in normal cognitive individuals (mean age 70.2 ± 5.7 years) was associated with more global Aβ PET deposition (cerebral amyloidosis) after APOE4 adjustment, while no correlation was found for total cholesterol, HDL-C or LDL-C. However, in this study amyloid-beta deposition in medial temporal, occipital, and basal ganglia regions was not positively associated with high serum TG [42].

The results of another longitudinal cohort study of 318 cognitively normal individuals concluded that increased midlife triglycerides (mean age 54 years) were associated with abnormal CSF Aβ42 together with CSF Aβ42/p-tau ratio 20 years later, after adjusting for multiple vascular factors, education, age, and APOE4 [44].

Data from Framingham Heart Study outline the influence of hypertriglyceridemia (≥287 mg/dL for males and ≥226 mg/dL for females) during midlife (40–60 years of age) on late-onset AD risk in APOE e4 negative participants, after adjustment for systolic blood pressure, and based on genetic markers. This longitudinal study with over 10 years of follow-up included 157 cases and 2882 controls with AD status and genotypes available [46]. This information can be used to tailor preventive strategies targeting triglyceride levels.

However, evidence is still controversial regarding the association of increased serum TG levels and cerebral amyloidosis or AD. A small cross-sectional study performed on 74 individuals (3 with mild dementia, 38 with mild cognitive impairment, and 33 clinically normal) did not find a correlation between TG or total cholesterol levels and cerebral Aβ quantified by Global PIB Index. Still, there was an independent association between LDL cholesterol, HDL cholesterol, and amyloid brain deposition [47].

Proitsi et al. found no difference between AD patients and controls regarding serum TG, total cholesterol, LDL, and HDL cholesterol in a subgroup of participants (102 cases versus 104 controls). However, they underlined the association of low-chain and very-low-chain triglycerides with AD in the untargeted lipidomic analysis on 142 AD and 152 control subjects [48]. The results of a large study using Mendelian randomization were negative regarding any causal association between lipid fractions and AD risk. The study included 17,008 participants with AD and 37,154 controls [49]. Evidence from cross-sectional small sample studies has shown a decreased TG level in dementia subjects [50,51]. The statistically significant values were only in Lepara et al. study that has outlined the presence of lower TG levels in probable AD individuals (24 females and 6 males) compared to control groups [50].

The relationship between TG and cerebral amyloidosis may be related to the circulating complex of amyloid β synthesized in enterocytes and triglyceride-rich lipoproteins that could disrupt the blood-brain barrier and subsequently increase brain amyloid deposition [52]. Furthermore, another possible pathway that leads to increased TG and amyloid-beta accumulation is the absence of peroxisome proliferator-activated receptor-gamma (PPAR-r) [53]. Studies on animals suggest that this receptor regulates adipose triglyceride lipase and facilitates the clearance of brain β amyloid [54].

In Table 2, we have summarized the main studies about triglycerides and cerebral amyloidosis.

### 3.3. Triglycerides and Vascular Cognitive Impairment

Vascular cognitive impairment (VCI) describes a large spectrum of cognitive dysfunction caused by vascular diseases including stroke, silent brain infarction, and subclinical brain injury [4]. The incidence and prevalence of VCI are age-related [55]. In individuals with dementia, there is frequently evidence of the coexistence of both vascular and AD lesions making it difficult to assess the exact contribution of each one to the cognitive decline [5,9,55].

In a study performed on 1143 subjects, the degree of cerebral atherosclerosis and arteriosclerosis was correlated with the severity of the cognitive impairment, including perceptual speed, which indicates a vascular cause, and episodic memory, which represents an important feature of AD [6]. The main incriminated mechanism of cognitive decline by cerebrovascular disease is represented by hypoperfusion [5]. Besides the well-known involvement of hypercholesterolemia in the atherosclerosis process, there is evidence that through endothelial dysfunction, foam cell formation, inflammation, and cytokines regulation, TG are participating in the atherosclerotic cardiovascular disease risk [19,20,21,22].

Studies have shown that the constellation of metabolic abnormalities represented by metabolic syndrome (MetS) is associated with a higher risk of vascular dementia but not AD [26,56,57]. In elderly Americans (≥60 years), MetS has a prevalence of 54.9 ± 1.7% [58].

Metabolic syndrome represents a cluster of conditions: abdominal obesity (waist circumference >40 inches in men and >35 inches in women), hyperglycemia (>100 mg/dL)/pharmacological treatment, high TG (≥150 mg/dL or pharmacological treatment), low HDL cholesterol (<40 mg/dL in men and <50 mg/dL—female or pharmacological treatment) and hypertension (>130/85 mmHg or pharmacological treatment). According to the National Cholesterol Education Program Adult Treatment Panel III (NCEP ATP III) revised criteria, MetS diagnostic requires at least three of these five cardiometabolic parameters [59]. It was found that MetS may increase vascular dementia risk, but the exact mechanism is not known. A possible explanation could be related to the microvascular damage which leads to white matter deterioration and disturbance of neuronal connectivity [57]. Several studies outlined the risk of cognitive impairment in people with white matter lesions [60,61].

MetS increases the risk of leukoaraiosis [62], silent brain infarction [63], and clinical stroke [64]. Impaired fasting glucose and hypertriglyceridemia were associated with leukoaraiosis independently of elevated blood pressure in a Japanese population with 1030 healthy subjects (mean age 52.7 years; 534 men and 496 women) [62]. However, impaired glucose tolerance alone was found to be an equal predictor of stroke events in elderly individuals as MetS [64]. MetS was found to be significantly associated with silent brain infarction, a predictor of both clinical overt stroke and dementia [63].

Regarding the risk of vascular dementia and MetS components, Solfrizzi et al. observed a synergistic effect with statistical significance of all items compared to the additive risk of individual components. The link was even stronger in subjects with high inflammation status and after the exclusion of undernourished participants. The study was performed on 2097 participants with 3.5 years of follow-up [57]. On the other hand, Raffaitin et al. demonstrated an independent association of each component of MetS with the risk of dementia. The cohort was a subsample of the Three-City Study and included 7087 participants aged ≥65 years with 4 years of follow-up [56]. High TG level was the only component of metabolic syndrome that was significantly associated with the incidence of all-cause and vascular dementia, while diabetes, but not impaired fasting glycemia, was significantly associated with all-cause and vascular dementia [56,65].

Bowler emphasizes the importance of recognizing cardiovascular risk factors, comprising all components of MetS, as risk factors for vascular dementia [66]. Considering single components of MetS and the risk of silent brain infarction, a study with 1588 healthy subjects determined that only high blood pressure and impaired fasting glucose had a strong significance, while high TG and low HDL levels showed marginal significance [63].

A cross-sectional study performed on 202 non-demented participants of the Biomarker Development for Postoperative Cognitive Impairment in the Elderly (BioCog study, patients aged 65-87 years with elective surgery in centres from Utrecht, the Netherlands, and Berlin, Germany) supports the idea that high TG level increases twice the possibility of cognitive impairment. However, this association was no longer statistically significant after adjustment for cerebrovascular and coronary heart disease [67]. A recent cross-sectional study with 108 participants aged ≥60 years with memory complaints from the Australian Imaging Biomarkers and Lifestyle (AIBL) study showed a positive correlation of the multiple risk factors in MetS with lower executive function and global cognitive performance [68].

No association was found between MetS and AD [56,57]. In contrast, a cross-sectional study performed on a random population-based sample of 980 individuals has shown that the prevalence of AD was higher in women with MetS, but not in men. Furthermore, over 80% of women with AD had MetS [69].

Table 3 summarizes the main studies about triglycerides and the vascular cognitive impairment.

## 4. Conclusions

Summarizing, the evidence of the TG level involvement in cognitive decline is still scarce and debatable.

High triglycerides level is associated with cognitive impairment, especially in large longitudinal studies, although there are several studies, the majority of them cross-sectional, that have shown no correlation or, on the contrary, even a reduction in the cognitive risk. Regarding data about TG-cerebral amyloidosis relationship, two longitudinal studies have correlated increased TG during midlife with abnormal CSF Aβ42 or late-onset AD after 20 years and 10 years, respectively. Furthermore, a cross-sectional study performed on normal cognitive elderly participants found an association of high serum TG with more global Aβ PET deposition. However, the data from AD patients did not show a correlation with elevated TG. The evidence might connect MetS to vascular cognitive impairment. Moreover, in one longitudinal study hypertriglyceridemia was found to be the only component of the MetS significantly associated with vascular dementia. However, more studies are necessary to clarify the role of TG and the other components of the MetS, possible confounders.

Considering that the management of hypertriglyceridemia benefits from lifestyle modification, diet, and specific drug therapy, future longitudinal studies are requested to appraise the triglycerides–cognitive impairment relationship.

## Figures and Tables

**Table 1 nutrients-13-02118-t001:** Characteristics of the clinical studies concerning serum triglycerides and cognitive decline.

Study, Year	Participants	Outcome	Results	Conclusions
Power M.C. et al. [7], 2018	13,997 participants of the Atherosclerosis Risk in Communities study	The association between measured serum lipids in midlife and subsequent 20-year change in performance on three cognitive tests	Elevated Chol, LDL-C, and TG were associated with greater 20-year decline on a test of executive function, sustained attention, and processing speed.	Elevated Chol, LDL-C, and TG in midlife were associated with greater 20-year cognitive decline.
Reynolds C.A. et al. [27], 2010	819 adults from the Swedish Adoption Twin Study of Aging aged 50 and older at first cognitive testing	The effect of lipids and lipoproteins on longitudinal cognitive performance and cognitive health in late life, 16 years follow-up	In women, higher HDL-C and lower apoB and TG predicted better maintenance of cognitive abilities than age. Lipid values were less predictive of cognitive trajectories in men.	High lipid levels may constitute a more important risk factor for cognitive health before age 65 than after.
Ancelin M.L. et al. [28], 2013	7053 community-dwelling elderly	The association between lipids and incident dementia, 7 years follow-up	In men without vascular pathologies, an increased incidence of all-cause dementia but AD was associated with high TG and low HDL-C levels. In women without vascular pathologies, low TG levels were associated with a decreased risk of AD.	Low HDL-C and high TG levels may be risk factors of dementia in elderly men whereas low TG is associated with decreased incident AD in women.
Henderson V.W. et al. [29], 2003	326 women in the Melbourne Women’s Midlife Health Project aged 52–63 years	The relation between serum lipids and memory in a healthy middle age cohort of women, 8 years follow-up	HDL-C and TG concentrations were unassociated with memory.	TG serum levels are not associated with better memory in healthy middle age women.
Lv Y.B. et al. [30], 2019	930 (mean age = 94.0 years) Chinese	The relationship of serum TG with cognitive function, activities of daily living (ADL), frailty, and mortality among the oldest old, 5 years follow-up	Each 1-mmol/L increase in TG was associated with a nearly 20% lower risk of cognitive decline, ADL decline, and frailty aggravation. Higher TG was associated with lower 5-year all-cause mortality. Nonelevated TG (less than 2.26 mmol/L) were associated with higher mortality risk.	In the oldest old, a higher concentration of TGs was associated with a lower risk of cognitive decline, ADL decline, frailty aggravation, and mortality.
He Q. et al. [31], 2016	112 MCI cases and 115 cognitively normal controls	The association of plasma lipids/lipoproteins with MCI	Plasma TG level was negatively associated with the risk of MCI. The adjusted odds ratio (OR) of MCI was significantly reduced for the highest quartile of plasma TG level.	Elevated plasma HDL and triglyceride were associated with the occurrence of MCI.
Sims R.C. et al. [32], 2008	121 African Americans adults	The relationship between elevated blood pressure and elevated TG, and verbal learning in a community-based sample of African Americans	TG levels were inversely related to California Verbal Learning Test-II performance. Higher TG levels were associated with poorer immediate, short delay and long delay recall.	TG levels may be related to cognitive functioning.
Parthasarathy V. et al. [33], 2017	251 nondemented elderly adults (54% male)	Association of increased TG with decreased executive function and memory in nondemented elderly subjects	TG levels were inversely correlated with executive function, but there was no relationship with memory.	TG levels are inversely correlated with executive function in nondemented elderly adults.
Yin Z.X. et al. [34], 2012	836 subjects aged 80 and older	The relationship between blood lipids/lipoproteins and cognitive function in the Chinese oldest-old	TG level was significantly negatively associated with cognitive impairment.	High normal plasma TG was associated with preservation of cognitive function.
Zhao B. et al. [35], 2019	1762 participants (aged 40–85)	The relationship between serum lipids and cognitive impairment	High serum TG was negatively associated with cognitive impairment in the middle-aged (≤55) male participants.	High serum TG may be protector of cognitive impairment in the middle-aged male participants.
Van Exel E. et al. [36], 2002	561 subjects 85 years old	The association between total and fractionated cholesterol and cognitive impairment	No differences in MMSE scores were found for TG.	No correlation between TG and cognitive decline.
Buyo M. et al. [37], 2020	2150 subjects aged between 60 and 90 years	The effect of metabolic syndrome and its components on global cognitive function	The score of MMSE was significantly negatively associated with TG in males.	In community-dwelling non-demented Japanese older adults, attention but not global cognitive function may be impaired by metabolic syndrome.
Bernath M.M. et al. [23], 2020	689 participants from the Alzheimer’s Disease Neuroimaging Initiative cohort, including 190 cognitively normal older adults, 339 with MCI, and 160 with AD	The association of TG with AD and the amyloid, tau, neurodegeneration, and cerebrovascular disease biomarkers for AD	PUTGs were significantly associated with MCI and AD.	PUTG component scores were significantly associated with diagnostic group and AD biomarkers.

Chol, total cholesterol; LDL-C, low-density lipoprotein cholesterol; TG, triglycerides; apoB, apolipoprotein B; HDL-C, high-density lipoprotein cholesterol; AD, Alzheimer disease; MCI, mild cognitive impairment; MMSE, Mini Mental State Examination; PUTG, polyunsaturated fatty acid-containing TGs.

**Table 2 nutrients-13-02118-t002:** Characteristics of some relevant clinical studies concerning triglycerides and cerebral amyloidosis.

Study, Year	Participants	Outcome	Results	Conclusions
Choi H.J. et al. [42], 2016	59 cognitively normal elderly individuals	The association between serum lipid measures and cerebral amyloid-beta (Aβ) deposition in cognitively normal elderly individuals.	Higher serum TG level was associated with heavier global cerebral Aβ deposition	Serum TG are closely associated with cerebral amyloidosis.
Nägga K. et al. [44], 2018	318 cognitively normal individuals	The effect of midlife lipid levels on Alzheimer brain pathology 20 years later in cognitively normal elderly individuals.	Higher levels of TG in midlife were independently associated with abnormal CSF Aβ42 and abnormal Aβ42/p-tau ratio. TG were also associated with abnormal Aβ PET in multivariable regression models.	Increased levels of triglycerides at midlife predict brain Aβ and tau pathology 20 years later in cognitively healthy individuals.
Peloso G.M. et al. [46], 2018	157 cases and 2882 controls, individuals 40–60 years old in the Framingham Heart Study	The interaction of a genetic risk score (GRS) of AD risk alleles with mid-life plasma lipid levels (LDL-C, HDL-C, and TG) on risk for AD.	There was a significant interaction between a GRS of AD loci and log TG levels on risk of clinical AD (*p* = 0.006).	Hypertriglyceridemia during midlife confers a higher risk of AD.

CSF, corticospinal fluid; Chol, total cholesterol; LDL-C, low-density lipoprotein cholesterol; TG, triglycerides; HDL-C, high-density lipoprotein cholesterol; AD, Alzheimer disease.

**Table 3 nutrients-13-02118-t003:** Characteristics of the clinical studies concerning triglycerides and vascular cognitive impairment.

Study, Year	Participants	Outcome	Results	Conclusions
Raffaitin C. et al. [56], 2009	7087 community-dwelling subjects aged > or =65 years recruited from the French Three-City cohort	Associations between MetS and its individual components with risk of incident dementia	High TG level was the only component of MetS that was significantly associated with the incidence of all-cause (hazard ratio 1.45 [95% CI 1.05–2.00]; *p* = 0.02) and vascular (2.27 [1.16–4.42]; *p* = 0.02) dementia.	The observed relation between high TG and vascular dementia emphasizes the need for detection and treatment of vascular risk factors in older individuals in order to prevent the likelihood of clinical dementia.
Kalmijn S. et al. [26], 2000	3555 men included in The Honolulu–Asia Aging Study	The long-term association between clustered metabolic cardiovascular risk factors measured at middle age and the risk of dementia in old age, 1965–1991	A higher cardiovascular metabolic risk factor burden in middle age increased the risk of dementia 25 years later.	Some individual risk factors, such as body weight, body fat distribution, and TG, were more strongly related to the risk of dementia than were other factors, and in particular of VaD.
Solfrizzi V. et al. [57], 2010	2097 participants (65–84 years old) from the Italian Longitudinal Study on Ageing	The relationship of MetS and its individual components with incident dementia, 3.5-year follow-up	MetS subjects compared with those without MetS had an elevated risk of VaD (adjusted HR, 3.82; 95% CI 1.32 to 11.06) associated with abnormal Aβ PET in multivariable regression models.	MetS subjects had an elevated risk of VaD that increased after excluding patients with baseline undernutrition and selecting MetS subjects with high inflammation.
Park K. et al. [62], 2007	1030 healthy persons (mean age 52.7 years) with no history of stroke	The relationship between LA and MetS in healthy subjects	MetS was significantly associated with the presence of LA (adjusted OR, 3.33; 95% CI, 2.30, 4.84). As for MetS components, elevated blood pressure, impaired fasting glucose and hypertriglyceridemia were independently associated with all grades of LA.	Hypertriglyceridemia was associated with LA independently of elevated blood pressure.
Feinkohl I. et al. [67], 2019	202 participants (aged 65 to 87 years) of the BioCog study	The role of MetS in cognitive impairment	Among the 5 MetS components, participants with elevated TG were at 2-fold increased odds of impairment (OR 2.09, 95% CI 1.08, 4.05, *p* = 0.028).	The finding was no longer statistically significant when cerebrovascular disease and coronary heart disease were additionally controlled for.

MetS, metabolic syndrome; TG, triglycerides; VaD, vascular dementia; LA, leukoaraiosis.

## Data Availability

Not applicable.

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
