# Peer review of "The Role of High Triglycerides Level in Predicting Cognitive Impairment: A Review of Current Evidence"

_nutrients, 2021, doi:10.3390/nu13062118_

Round 1

Reviewer 1 Report

Here the Authors review the evidence linking AD to dyslipidemia.  This could be a useful resource for the field, but it needs improvements, particularly where describing the outcomes from the studies discussed, and in the synergizing the topics presented.

  • L30 “Neurodegenerative and vascular mechanisms were most frequently 30 assessed in patients with dementia [3].” – This is not clear.
  • L33 – “incriminating”? - revise word choice
  • L36 -Relationship?
  • L 51 - The preclinical phase used 51 in research is based on biological criteria where the explorations for β amyloid and tau 52 protein are positive, with or without the presence of neurodegeneration, but in the ab-53 sence of symptoms [14]. – not clear what “used in research means”, the category perhaps.
  • Early part of the introduction is broken into too many paragraphs
  • L77 – reference needs repeating here.
  • Banks et al., L102 – showed that exogenous triolein can cross BBB – this needs to be clarified. e. that whether TG rich lipoproteins cross the BBB requires further study.
  • The section 3.1 showing both positive and negative associations with TGs and AD is both useful and informative to the field. However, it would be more useful if more information about the TGs (i.g. total TGs, vs VLDL-TGS) could be included so that we may start to interpret more about the different outcomes from these studies.
  • L156 – please specify the lipid species.
  • Each section requires a summary statement at least – one that leads into the next would be even better.
  • In reference 43 and 44, what are the dyslipidemia characteristics?
  • Again with reference 46 – what are the outcomes?
  • L209 – Is low chain, short chain?
  • The prose is split into too many paragraphs throughout the document.
  • There is little synergy of the data, and a table or summary would be useful.

Author Response

  • L30 “Neurodegenerative and vascular mechanisms were most frequently 30 assessed in patients with dementia [3].” – This is not clear.

 Response: We have replaced the phrase with „Two of the commonest subtypes in order of frequency are Alzheimer’s Disease and vascular dementia, but it is difficult, particularly in epidemiological studies, to establish an accurate subtype diagnosis” to underline the distinctive mechanisms – neurodegeneration and vascular – in the etiology of dementia.

  • L33 – “incriminating”? - revise word choice

 Response: We have replaced the word and rephrased the sentence with „The accumulated findings of a causal role of dyslipidemia in the etiology of neurodegeneration and cognitive decline are strong, but the results are still debatable.”

 L36 -Relationship?

 Response: The word „relation” was replaced with „relationship”.

  • L 51 - The preclinical phase used 51 in research is based on biological criteria where the explorations for β amyloid and tau 52 protein are positive, with or without the presence of neurodegeneration, but in the ab-53 sence of symptoms [14]. – not clear what “used in research means”, the category perhaps.

 Response: We have completed the phrase with a detail between brackets: The preclinical phase used in research settings (interventional and observational studies).

  • Early part of the introduction is broken into too many paragraphs.

 Response: We have contracted two paragraphs corresponding to lines 50-51 and 61-62, respectively.

 L77 – reference needs repeating here.

 Response: Reference 17 was repeated accordingly.

 Banks et al., L102 – showed that exogenous triolein can cross BBB – this needs to be clarified. e. that whether TG-rich lipoproteins cross the BBB requires further study.

 Response: We have rephrased and offered more explanation in „Banks et al. detected TG in human cerebrospinal fluid and moreover, they found that radioactive TG triolein crosses the blood-brain barrier (BBB) in mice. The authors have demonstrated the improvement of cognition after gemfibrozil administration with a subsequent decrease of serum TG level [24]. However, the evidence regarding the passage of the BBB by the TG is very scarce and requires more studies”.

  • The section 3.1 showing both positive and negative associations with TGs and AD is both useful and informative to the field. However, it would be more useful if more information about the TGs (i.g. total TGs, vs VLDL-TGS) could be included so that we may start to interpret more about the different outcomes from these studies.

L156 – please specify the lipid species.

 Response: We have completed the information with „principal component 3 (PC3) and principal component 5 (PC5), respectively”.

  • Each section requires a summary statement at least – one that leads into the next would be even better.

 Response: We summarized each section by adding a table with the most relevant studies.

  • In reference 43 and 44, what are the dyslipidemia characteristics?

 Response: Reference 43 – total cholesterol (information was added in the text – line 184). Reference 44 – triglycerides – line 196.

 Again with reference 46 – what are the outcomes?

 Response: Explanation inserted between brackets - hypertriglyceridemia (≥287 mg/dl for males and ≥ 226 mg/dl for females), followed by a short conclusion of the study.

  • L209 – Is low chain, short chain?

 Response: Not specified

 The prose is split into too many paragraphs throughout the document.

 Response: We have tried to compress the data in fewer paragraphs.

 There is little synergy of the data, and a table or summary would be useful.

 Response: Every section ends with a Table summarizing important data.

Reviewer 2 Report

In this review, it mainly described the correlation of triglycerides (TGs) with cognitive decline, cerebral amyloidosis and vascular impairment. In each section, much clinical data was used to support the thesis and we have better learned the relationship between TGs and cognitive decline, cerebral amyloidosis, and vascular cognitive impairment, meanwhile, it revealed the role of TGs involving in blood brain barrier crossing and amyloid production, and then we have a deeper understand the function of TGs in regulating the learning and memory. In this paper, the author also described the debatable part of the TGs level involvement in cognitive decline, which indicated the role of TGs showed aging-dependent, gender-dependent or disease-dependent differences. Then we know that the TGs level is associated with many diseases, especially the neurodegenerative disease, and we can change the lifestyle and diet to get the TGs level under control.

Minor:

A lot of clinical data (ages, sex, symptom) were shown in each section. The author needs to present these data in several tables.

Author Response

We summarized each section by adding a table with the most relevant studies.